# Adult-Onset Still’s Disease—A Complex Disease, a Challenging Treatment

**DOI:** 10.3390/ijms232112810

**Published:** 2022-10-24

**Authors:** Luana Andreea Macovei, Alexandra Burlui, Ioana Bratoiu, Ciprian Rezus, Anca Cardoneanu, Patricia Richter, Andreea Szalontay, Elena Rezus

**Affiliations:** 1Department of Rheumatology, “Grigore T. Popa” University of Medicine and Pharmacy, 16 Universitatii Street, 700115 Iasi, Romania; 2Clinical Rehabilitation Hospital, 700661 Iasi, Romania; 3Department of Internal Medicine, “Grigore T. Popa” University of Medicine and Pharmacy, 700115 Iasi, Romania; 4III Internal Medicine Clinic, “St. Spiridon” County Emergency Clinical Hospital, 700111 Iasi, Romania; 5Department of Psychiatry, “Grigore T. Popa” University of Medicine and Pharmacy, 700115 Iasi, Romania; 6Institute of Psychiatry “Socola”, 700282 Iasi, Romania

**Keywords:** Adult-onset Still disease, neutrophil activation, pro-inflammatory cascade, anti-cytokine therapy

## Abstract

Adult-onset Still’s disease (AOSD) is a systemic inflammatory disorder with an unknown cause characterized by high-spiking fever, lymphadenopathy, hepatosplenomegaly, hyperferritinemia, and leukocytosis. The clinical course can be divided into three significant patterns, each with a different prognosis: Self-limited or monophasic, intermittent or polycyclic systemic, and chronic articular. Two criteria sets have been validated. The Yamaguchi criteria are the most generally used, although the Fautrel criteria offer the benefit of adding ferritin and glycosylated ferritin values. AOSD’s pathogenesis is not yet completely understood. Chemokines and pro-inflammatory cytokines, including interferon (IFN)-γ, tumor necrosis factor α (TNFα), interleukin (IL)-1, IL-6, IL-8, and IL-18, play a crucial role in the progression of illness, resulting in the development of innovative targeted therapeutics. There are no treatment guidelines for AOSD due to its rarity, absence of controlled research, and lack of a standard definition for remission and therapy objectives. Non-steroidal anti-inflammatory drugs (NSAIDs), corticosteroids (CS), and conventional synthetic disease-modifying antirheumatic drugs (csDMARDs) are used in AOSD treatment. Biological therapy, including IL-1, IL-6, IL-18, and IL-17 inhibitors, as well as TNFα or Janus-kinases (JAKs) inhibitors, is administered to patients who do not react to CS and csDMARDs or achieve an inadequate response.

## 1. Introduction

Adult-onset Still’s disease (AOSD) is a rare inflammatory disease with an unknown cause [1]. Bywaters, a London doctor, first used the term AOSD in the medical literature in 1971 when he described the condition in a small group of 14 patients ranging in age from 17 to 35 years [2].

Its incidence ranges between 0.16 and 0.4 per 100,000 people, depending on the population studied [3,4]. Although the majority of cases present between the ages of 16 and 35, with a slight female predominance, reports of older patients with AOSD are on the rise [5]. 

The most prevalent clinical signs of the disease [6] include high-spiking fever, arthritis, and a transient salmon-pink maculopapular rash. Odynophagia and occasionally pharyngitis are symptoms that accompany fever [7]. Additionally, increased liver enzymes, lymphadenopathy, hepatosplenomegaly, hyperferritinemia, and white-blood-cell count (WBC) of 10,000/mm^3^, primarily neutrophilic polymorphonuclear (PMNs) cells, are frequently detected and supportive of the diagnosis [8,9]. Myalgia is common, although myositis and polymyositis are rare [1]. Patients with AOSD are also likely to have elevated levels of inflammatory markers, including elevated levels of the C-reactive protein (CRP) and erythrocyte sedimentation rate (ESR) [10].

## 2. AOSD-Clinical Picture and Diagnostic Criteria

Clinical manifestation is highly variable [11], making diagnosis difficult. The disease has no clinical, biological, histological, or radiological hallmark. This lack of specificity frequently results in a missed or overdue diagnosis. A delay in diagnosis has been shown to influence the future response to therapy. According to Italian and French studies, the time between the onset of symptoms and the final diagnosis of AOSD ranges from 1.5 to 4 years [12]. Failure to obtain a rapid diagnosis of AOSD leads to a chronic disease course, as demonstrated by Kalyoncu et al. [13].

Although they are primarily designed for research, most physicians use the Yamaguchi [9] and Fautrel [9,14] classification criteria for AOSD in practice. These two are the most sensitive and specific diagnostic criteria [15]. They include exclusion criteria with high sensitivity and specificity, such as infections, malignancies, and other autoimmune diseases [16,17,18].

Generally, the Yamaguchi criteria are separated into major and minor criteria. These criteria can only be applied if exclusion criteria have been carefully considered. To classify and diagnose AOSD using the Yamaguchi criteria, you must have at least five criteria, with no less than two being major and no exclusion criteria. Fautrel diagnostic criteria include both major and minor criteria but no exclusion criteria; they do, however, include some recently described serologic criteria, such as serum ferritin. Four major criteria or three major criteria and two minor criteria are necessary for a positive diagnosis [9,19]. Figure 1 and Figure 2 below highlight the two sets of criteria used in practice.

The clinical course of AOSD has been classified into three distinct phenotypes based on the evolution of symptoms over time: Self-limiting or monophasic, intermittent or polycyclic, and chronic evolution, each with a different prognostic significance [1,12,20,21,22]. Figure 3 describes these specific patterns.

Maria et al. [23,24] proposed a dichotomous perspective of AOSD, differentiating disease subtypes based on the dominant clinical manifestation. Thus, two subsets of patients can be distinguished: Those with predominant systemic clinical features, such as fever and skin rash, and those with principal articular involvement, similar to classical rheumatoid arthritis (RA). 

Predictive factors for each subset’s evolution have been identified. Female gender, proximal arthritis at disease onset, thrombocytosis, and steroid dependence represent the articular pattern [25]. The systemic subset appears to be associated with high fever (>39 °C), high levels of liver enzymes or acute phase reactants, thrombocytopenia, and hyperferritinemia [25,26]. Interleukin (IL)-18, interferon (IFN)-γ, IL-10, and IL-4 are related to systemic AOSD, whereas IL-6, IL-17, and IL-23 are associated with arthritic AOSD. However, cytokine dosage is not routinely measured [26,27].

## 3. AOSD-Prognosis and Complications

The prognosis for AOSD is favorable, with an estimated specific mortality rate of 1–3% [17]. Some patients, however, experience complications. Even so, early diagnosis and prognosis assessment may help to reduce the disease’s critical problems, such as macrophage activation syndrome (MAS), thrombotic thrombocytopenic purpura, respiratory distress syndrome, and diffuse alveolar hemorrhage [8]. The disease’s multi-visceral involvement may significantly reduce AOSD patients’ life expectancy.

The most severe complication of AOSD is MAS. The prevalence ranges between 10 and 15% and is associated with a high mortality rate [12]. Infections or medications, in combination with uncontrolled and prolonged inflammation in patients with a genetic predisposition, may cause this potentially fatal condition [17,18]. MAS can occur either at the time of diagnosis or later on. Specific predictive or diagnostic factors are lacking [19]. High fever, hepatosplenomegaly, cytopenias, coagulopathy, extreme hyperferritinemia, and hemophagocytosis on bone marrow aspirates are the most common symptoms of MAS.

## 4. AOSD-Pathogenesis

The pathophysiology of AOSD is still unknown. However, factors such as an imbalance in innate and adaptive immunity and increased inflammatory cytokines contribute to disease development [28].

There is no evidence that family aggregation plays a role in the occurrence of AOSD. Still, previous research has found a link between genetic susceptibility and human leukocyte antigens (HLA) gene polymorphisms, including HLA-Bw35, -B17, -B18, -B35, -DR2, -DR4, -DR5, -DQ1, -DRw6, -DRB1, and -DQB1 [29]. In a Chinese multicenter cohort consisting of 264 AOSD cases and 2420 controls, the first genome-wide association research was performed to evaluate genetic variables determining susceptibility to AOSD. This analysis identified both HLA class I and class II regions as susceptibility loci for AOSD [30].

Although multiple HLA alleles have been linked to a predisposition to the disease, a trigger is still required to set off the chain reaction of inflammation.

Some clinical signs of AOSD, such as fever spikes, lymphadenitis, and elevated liver enzymes, resemble viral or bacterial infections, suggesting that infection may initiate the inflammatory response in AOSD. Adenovirus, Human immunodeficiency virus (HIV), *Mycoplasma pneumoniae*, parvovirus B19, Epstein-Barr virus (EBV), rubella virus, measles morbillivirus (MeV), hepatitis virus, influenza virus, rubella, and severe acute respiratory syndrome coronavirus 2 (SARS-CoV-2), emerging in late 2019, are among the reported infectious triggers [3,8].

The disease’s specific pathogenic pathways are still partially known. Neutrophil activation, a defining feature of the pathophysiology of AOSD, is responsible for the onset and progression of inflammation by releasing a vast array of granular enzymes and antimicrobial proteins. During an acute flare of the disease, over 80% of patients show neutrophilic leukocytosis, which helps distinguish AOSD from other rheumatic conditions [8].

Neutrophil and macrophage activation is a characteristic of AOSD, potentially due to pro-inflammatory IL-18 signaling. PMN CD64, a neutrophil activation marker, has recently been elevated in patients with active AOSD. As its expression typically reflects the degree of disease activity, intercellular adhesion molecule-1 (ICAM-1), elevated by IL-18, has also been suggested as a possible clinical marker [31]. Additionally, macrophage-colony stimulating factor (M-CSF), a markedly enhanced cytokine in severely AOSD patients, appears to be orchestrating the activation and differentiation of macrophages [32].

Neutrophil extracellular traps (NETs) are essential in the intensive activation of macrophages and the stimulation of the overproduction of many pro-inflammatory cytokines [33]. Histones are the predominant protein component of NETs, followed by granule-derived peptides and enzymes, including neutrophilic elastase, myeloperoxidase, calprotectin, cathepsin G, leukocyte proteinase 3, lysozyme C, and neutrophil defensins [34]. In AOSD, the S100 protein is the most commonly explored field connected to NETs. Calprotectin is defined as a combination of two calcium-binding proteins of the S100 protein family, S100A8 and S100A9. Patients with AOSD have higher levels of the S100 family of proteins, produced mainly by neutrophils and macrophages, than healthy controls. They function as ligands of Toll-like Receptor (TLR) 4 or the receptor for advanced glycation-end products (RAGE) to accelerate neutrophils and trigger the production of pro-inflammatory cytokines [35]. Activated neutrophils and macrophages release calprotectin and the macrophage migration inhibitory factor (MIF), which are excellent indicators of disease activity and severity [31,36].

The pro-inflammatory cascade is most likely initiated by danger signals such as pathogen-associated molecular patterns (PAMPs) or damage-associated molecular patterns (DAMPs). Pattern recognition receptors (PRRs), TLRs, activate the Nucleotide-Binding Domain, the Leucine-Rich-Containing Family, and Pyrin Domain-Containing-3 (NLRP3) inflammasomes [37]. Caspase enzymes stimulate the generation of IL-1β and IL-18, the characteristic cytokines of active AOSD [38]. IL-1β and IL-18 also stimulate the release of pro-inflammatory cytokines such as IL-6, IL-8, IL-17A, and tumor necrosis factor (TNF)α [39,40]. IL-1β can further activate macrophages, which are essential players in the cytokine storm or MAS [18,41]. IL-1β and IL-18 are thought to be primarily responsible for systemic symptoms such as fever and general weariness, whereas TNFα is responsible for arthritis and IL-6 is intermediate. 

IL-1β is the main pro-inflammatory cytokine. It is neither present nor detectable by routine immunoassays in healthy tissues but is generated during inflammation by myeloid cells (blood monocytes, tissue macrophages, and dendritic cells). Exogenous TLR agonists or endogenous cytokines such as TNFα [42] or IL-1β itself boost production. This self-sustaining activation of IL-1β is an autoinflammation-causing mechanism.

IL-1β is released in the extracellular area [42], where it produces a variety of pro-inflammatory effects. These include nuclear factor-kB (NF-kB), activator protein-1 (AP-1), c-Jun N-terminal kinase (JNK), and other mitogen-associated protein kinases (MAPKs) linked to immunological responses [42,43]. These signaling cascades activate other mediators, most notably IL-6, IL-2, interferons, chemokines, prostaglandins, and endothelial adhesion molecules [44].

Increasing concentrations of IL-1β cause the most important AOSD symptoms, including fever, an increase in acute-phase reactants, neutrophilia, rash, musculoskeletal discomfort, hepatosplenomegaly and lymphadenopathies, serositis, hypotension, and shock.

IL-1β is also a critical cytokine in the promotion of adaptive immunity. IL-1β promotes the development of CD4+ T cells into proinflammatory T cell populations such as T helper (Th)1 and Th17 cells, and it can also drive the proliferation and differentiation of antigen-specific CD8+ T cells [45].

Kudela et al. [46], in their investigation, discovered a considerable increase in IL-18 blood levels in active AOSD compared to other rheumatic diseases, as well as a strong connection between IL-18 serum levels and disease activity in AOSD. These findings indicate IL-18’s potential function as an important biomarker in AOSD. IL-18 has also been associated with serum ferritin, CRP, and neutrophil count [31,47].

In contrast to other inflammatory conditions, including RA, systemic lupus erythematosus (SLE), ankylosing spondylitis (AS), and psoriatic arthritis (PsA), the circulation amount of free IL-18 is higher in individuals with AOSD during both the active and inactive phases of the condition. Those with active illness had greater levels of free IL-18 than patients with inactive disease, suggesting that IL-18 may be used as a biomarker to assess the disease activity of AOSD [38,48]. AOSD patients with MAS are also observed to have high levels of IL-18 [49].

IL-8 is a proinflammatory cytokine that mobilizes, stimulates, and degranulates neutrophils at the site of inflammation [31]. It is primarily generated by activated macrophages and serves as a chemotactic agent of inflammatory cells. Chen et al. [37] discovered that the blood IL-8 level was a significant predictor of chronic arthritis. 

Serum IL-17 pro-inflammatory cytokine levels were more significant in AOSD patients and linked to Th17-circulating cells. Th17 cells and IL-17 levels were lowered after therapy administration, suggesting that Th17-targeted therapies may have a therapeutic effect in managing those disorders [50].

The role of IFN-γ in AOSD is still debated. It stimulates pro-inflammatory responses such as host defense responses and regulatory functions such as neutrophil-specific chemokine inhibition and T-cell apoptosis induction. CXC motif chemokine 10 (CXCL10) and IL-18 are two examples of IFN-γ-induced cytokines or chemokines that are considerably elevated in AOSD [51]. CXCL10 levels were shown to be higher in AOSD patients compared to those of RA patients and healthy controls and linked to AOSD disease activity indices, as described by Han et al. [52].

Han et al. [52] indicated that blood levels of CXCL9, CXCL10, and CXCL11 in patients with AOSD were linked with many inflammatory markers and systemic scores. In contrast, they decreased due to treatment-induced improvement in disease activity.

In addition to enhancing immune response and inflammatory processes, IL-6 contributes to developing AOSD [1,28,53]. As a proinflammatory cytokine, IL-6 may be responsible for fever, rash, and the synthesis of acute-phase proteins in AOSD [37]. Skin lesional biopsies from those with the distinctive salmon-colored rash demonstrated elevated IL-6 levels. Furthermore, IL-6 may contribute to elevated ferritin levels by stimulating its synthesis by the liver, along with CRP and other acute-phase proteins [31].

Patients with AOSD had elevated TNFα levels in their sera and tissues compared to healthy controls, regardless of disease activity. On the other hand, blood levels of soluble tumor necrosis factor-receptor-2 (sTNF-R2) were associated with serum CRP levels, indicating its potential utility as an indicator of disease activity [30].

Ferritin is widely documented as a common AOSD mediator [54,55]. Ferritin production can be increased in response to inflammatory cytokines such as IL-1β and IL-6. Furthermore, ferritin can trigger inflammatory pathways to exacerbate the inflammatory process, lending to the idea that ferritin is more than just an observer in acute-phase reactions [56].

In addition, dysfunctional natural killer (NK) cells, elevated Th1 and Th17 cells, increased IFN-γ and IL-17 levels, and various alarmins, such as the S100 proteins, contribute to the pro-inflammatory environment that promotes aberrant human immune system responses [50]. NK cells facilitate the inflammatory cascade. Immunological responses rely on NK cell cytolytic activity to eliminate infections and preserve lymphoid and myeloid immune homeostasis. A lack of cytotoxicity will result in chronic lymphocyte and macrophage activation [57,58]. Even though the cytolytic activity of NK cells is diminished in AOSD, the ability to secrete IFN-γ is increased due to the overexpression of IL-12 and IL-15 receptors on these cells [59].

Apart from intensified inflammatory cascade, it is theorized that inadequate resolution of inflammation contributes to the “cytokine storm” of AOSD. Active AOSD is associated with increased anti-inflammatory cytokines IL-10 and IL-37, which may attenuate the aberrant inflammatory response [60]. IL-10 may block macrophage activation, restrict neutrophil migration, and regulate the production of IL-1, IL-6, and TNFα [61]. IL-37 also may impede the production of IL-1, IL-6, and TNFα [62,63]. In addition, IL-10 and IL-37 can stimulate the polarization of anti-inflammatory macrophages, which aids in resolving an excessive inflammatory response [64].

Figure 4 summarizes the aspects related to the pathogenesis of this condition.

## 5. AOSD-Treatment

### 5.1. Goals and Categories of Therapy

The primary objectives of treatment for AOSD are to reduce inflammation and promote the resolution of systemic and articular symptoms and prevent organ damage and MAS [65]. Nevertheless, many therapy choices, such as non-steroidal anti-inflammatory drugs (NSAIDs), corticosteroids (CS), and conventional synthetic disease-modifying antirheumatic drugs (csDMARDs) may only be appropriate for people with moderate, frequently self-limiting disease. In life-threatening instances and steroid-dependent patients, supplemental therapy with a second-line medication is required [24].

There are currently no internationally accepted AOSD management recommendations. Few country-specific guidelines have been published for the management of AOSD, and a treat-to-target strategy is still absent. In 2017, the Japanese Ministry of Health established AOSD’s first clinical practice guidelines [66]. For AOSD patients with significant organ involvement, they proposed systemic glucocorticoids for relieving clinical symptoms with high-dose intravenous pulse glucocorticoid treatment. Methotrexate (MTX) was strongly suggested for individuals with steroid resistance [66,67].

Recommendations for the use of IL-1 inhibitors in the treatment of AOSD were developed by an Italian expert group in 2019. A large percentage of patients achieved quick and persistent remission of systemic symptoms and normalized inflammatory markers, which led the panel to conclude that the research consistently established the positive effect of IL-1 inhibitors [23,67].

Because of the condition’s rarity, there are few well-designed prospective studies on AOSD patients and even fewer randomized controlled trials examining management methods in AOSD patients. NSAIDs and CS are used as first-line therapy for AOSD, followed by csDMARDs in steroid-refractory individuals and biologics in those resistant to conventional treatment [42]. The efficacy of these treatments was inadequate, particularly in severe symptoms. Standard methods cannot manage disease in at least 30–40% of patients, indicating a significant medical need for specific therapy [68]. Treatment should be initiated immediately so that the damaging inflammatory process may be stopped while it is still in its early, reversible phases [69].

The introduction of diverse biologic medication in refractory AOSD was encouraged by successful discoveries in treating other chronic rheumatic disorders. Because of their roles in generating and exacerbating a destructive systemic inflammatory response, IL-1, IL-6, IL-18, and TNFα are essential mediators in AOSD pathophysiology [30,68]. Consistent with these data, cytokine-blocking medicines emerged as the most appropriate and acceptable treatments for AOSD, especially in difficult-to-treat and severe patients [70].

### 5.2. NSAIDs, CS and csDMARDs Treatment 

The primary objectives of treatment for AOSD are the resolution of systemic and articular symptoms and the prevention of organ damage and MAS [65].

Despite their low overall efficacy, CS and NSAIDs are invariably the first-line therapy for both clinical characteristics. For more severe cases, csDMARDs and biological DMARDs (bDMARDs) are available [71,72,73].

#### 5.2.1. NSAIDs

NSAIDs are the first-line treatment, especially in the absence of systemic symptoms [24]. Approximately twenty percent of patients with a moderate or self-limiting disease (e.g., low-grade fever, rash, arthralgia) achieve clinical control with NSAIDs. More than eighty percent of AOSD patients did not achieve remission with NSAIDs, and nearly twenty percent experienced side effects [66].

When using NSAIDs for an extended period, it is essential to be aware of the potential adverse effects, including gastrointestinal bleeding and the possibility of developing renal or hepatic insufficiency [67]. 

#### 5.2.2. CS

The dosage of CS therapy depends on the severity of the disease: Prednisone at 0.5 to 1 mg/kg per day is the typical starting regimen. At the same time, high-dose intravenous pulse glucocorticoids are indicated for patients with life-threatening internal organ involvement (such as severe hepatic involvement and MAS) [74]. The reaction is frequently rapid and long-lasting for both articular and systemic problems.

Methylprednisolone may be ineffective in managing the most severe symptoms, particularly in people with resistant forms or MAS characteristics. Other CS, such as dexamethasone, should be tried in such circumstances [75].

Because of side effects, such as gastrointestinal bleeding, hypertension, diabetes, tachycardia, cataract, osteoporosis, psychosis, and weight gain, prolonged use of steroid medicine should be avoided [42,76].

It is important to note that abrupt reductions in steroid daily intake may result in illness relapses; consequently, the posology should be reduced gradually [75].

NSAIDs and CS enhance the risk of cardiovascular disease [77]. As a result of their dose-dependent toxicity, they should be reduced gradually and stopped once remission is established. Several AOSD patients, however, experience illness flares or become steroid-dependent after tapering or interruption.

#### 5.2.3. csDMARDs

csDMARDs are commonly regarded as steroid-sparing medications and are frequently used with CS or NSAIDs to provide appropriate disease management.

There are no controlled data from randomized clinical studies on the effectiveness and safety of csDMARDs in treating AOSD. In observational investigations, csDMARDs were shown to produce remission in up to 80% of AOSD patients, making them good steroid-sparing treatments for resistant and severe AOSD cases. The majority of AOSD patients were given at least one csDMARDs, the most common being MTX. It is administered with a beginning dose of 7.5 to 15 mg once weekly, followed by a plan of dosing increases up to 25 mg/week [30].

MTX was reported to be efficacious for disease control in systemic and chronic articular AOSD in the research of Gerfaud-Valentin published in 2014 [3], notably in 40–70% of steroid-dependent AOSD patients. The use of MTX in AOSD patients with liver damage is not an absolute contraindication, although constant transaminase monitoring is required [30,78].

MTX is the preferred option as first-line steroid-sparing therapy, despite studies showing the effectiveness of other traditional csDMARDs such as cyclosporine A, azathioprine (AZA), leflunomide (LEF), and hydroxychloroquine (HCQ) in AOSD patients [25].

For the therapy of the chronic articular pattern and MAS, MTX and cyclosporine A are most commonly used, respectively. However, data on AZA or LEF are rare and originate from small case series or case reports [28].

In addition, cyclosporine is the basis for treating MAS [79]. Nonetheless, these pharmacological medications are linked to a substantial burden of toxicity, which most commonly includes renal, gastrointestinal, hepatic, and hematological adverse responses [80,81].

Furthermore, 17–32% of AOSD patients with mainly severe clinical symptoms achieve only partial remission or are resistant to first-line CS and second-line csDMARDs [82]. These patients are referred to as “refractory AOSD” patients, and they frequently require larger CS dosages, longer treatment durations, and the concomitant introduction of bDMARDs [3,21].

### 5.3. IL-1 Inhibitors

As IL-1 is thought to play a role in the development of AOSD [31], and its ligands and receptors are released mainly through activated macrophages [83,84], inhibiting IL-1 with drugs appears to be a reasonable therapeutic strategy for AOSD patients.

IL-1 inhibitors are rapidly effective in managing many clinical and laboratory indications of AOSD, particularly in patients with the disease’s systemic form [70]. Anti-IL-1 therapy is also successful in individuals with AOSD who have failed conventional treatment [23,24,85].

The rapid and sustained response to IL-1 inhibitors enables patients to use less CS medication [22,23]. The efficiency of IL-1 inhibitors in treating AOSD varied across studies, ranging from 50 to 100 percent, and the percentage of remission varied between 22 and 100 percent (median 70 percent) [23]. Additionally, IL-1 inhibitors have a good safety profile. 

There are currently three IL-1 inhibitors available for AOSD: Anakinra (an IL-1R antagonist), canakinumab (an anti-IL-1 monoclonal antibody), and rilonacept (a soluble IL-1 trap molecule).

#### 5.3.1. Anakinra

Anakinra is a non-glycosylated version of the human IL-1 receptor antagonist (IL-1Ra), which binds to the IL-1 receptor (IL-1RI) and inhibits the activation of this receptor by IL-1α or IL-1β [44]. It was officially approved in 2001 for treating RA, but the first cases of its use in AOSD were not recorded until 2005 [86,87]. Anakinra [88] was the first biological molecule to demonstrate efficacy in treating systemic and articular signs of AOSD.

According to multiple trials, anakinra appears to be more effective when provided early in the disease. It was found to be more beneficial for patients with highly active systemic AOSD than those with isolated chronic arthritis [9].

In 2015, Ortiz-Sanjuan et al. reported that after one year of anakinra medication, the frequency of joint symptoms, cutaneous rash, fever, and ferritin serum levels decreased significantly in nearly 40 patients. In addition, an overall decline in daily steroid consumption was identified [89].

In a case report by Dall’Ara et al. [90], 13 patients with AOSD were treated with anakinra as first- or second-line biologic therapy, and 12 exhibited full remission after a median of 61 months of follow-up.

Colafrancesco et al. conducted extensive research in 2017 on 140 individuals with AOSD from 18 Italian centers. NSAIDs, CS, and csDMARDs were frequently mentioned as past therapy. All clinical and serological signs of AOSD improved rapidly with anakinra administration within the first three months of treatment [23].

Vercruysse et al. [91] concluded in their analysis of 15 patients treated with anakinra that two critical characteristics are linked to a considerable therapeutic response: A systemic form and the lack of arthritis.

In a 2020 randomized, placebo-controlled study by Schanberg et al. [92] evaluating the use of anakinra in 12 patients with Still’s disease (nine children and three adults, *n* = 6 placebo, *n* = 6 anakinra), six patients on anakinra demonstrated a rapid response at week 2, defined as the absence of fever and a 30% improvement in American College of Rheumatology (ACR) criteria 30 (ACR30).

Vitale et al. conducted a series of investigations in a multicenter study with 141 individuals with AOSD in Italy to assess early versus delayed anakinra treatment. Participants were randomized to different treatment groups based on disease length, anakinra therapy duration, the time between disease start and treatment beginning, and previous treatment regimens (mainly including CS, csDMARDs, and biologic agents). After 3, 6, and 12 months, there were no statistically significant changes in efficacy between groups. However, patients receiving anakinra as soon as AOSD onset may better manage systemic inflammation and articular symptoms [93].

A meta-analysis of nine clinical studies indicated that anakinra could reduce or even eliminate concurrent CS without AOSD flare-ups [94].

Bodard et al. [95] evaluated the efficacy of anakinra in 23 patients with AOSD in 2021, eight of whom had cardiac involvement including pericarditis and myocarditis with tamponade, and they reported positive outcomes in all of them.

After 24 months of treatment with anakinra, Campochiaro et al. [96] reported drug retention rates (DRR) of 53.1% in 41 patients with AOSD.

Regarding the safety of treatment with Anakinra, studies show a good profile. In retrospective research by Dall’Ara et al. [90] involving 18 patients, 13 of whom received anakinra, one patient with AOSD was found to have developed a drug-induced rash. In the study by Vitale et al., three patients experienced significant adverse effects (SAE): A 52-year-old man developed pneumonia after 17 months, a 65-year-old male developed lower limb ulcers after 110 months, and a 67-year-old male patient developed pneumonia after nine months of anakinra medication. Overall, the study authors found adverse events (AEs) and SAEs in 1.7% of patients (*n* = 72) treated with IL-1 blockers, confirming their acceptable safety profile [97].

In the extensive retrospective study by Colafrancesco et al., 47 of 140 patients reported AEs. The most common AEs were in situ or diffuse skin reactions and infections (three cases of urinary tract infections, three cases of pneumonia, and one case of recurring dental abscesses). The majority of affected patients were administered daily doses of 100 mg anakinra. In 75% of instances, medication was discontinued due to the persistence of severe skin responses during ongoing treatment. During a 35-month average follow-up period, leucopenia, thrombopenia, and lymphoproliferative diseases were also found [23].

Skin responses at the injection site are the most frequent and consistently reported AEs, according to Vastert et al.’s study of 27 trials. Notably, it is significant to point out that these AEs are mild to moderate in severity and usually disappear in 4 to 6 weeks without the need to stop using anakinra. According to the same analysis, people who have already experienced liver dysfunction are more likely to experience hepatotoxicity when taking anakinra [98].

In 2021, Campochiaro et al. reported injection site responses as the most common AEs, occurring in 4 of 41 patients with one case of zoster infection reactivation in a retrospective single-center cohort analysis [96].

#### 5.3.2. Canakinumab

Canakinumab is a fully human monoclonal antibody against IL-1 that prevents the production of inflammatory mediators by inhibiting downstream targets and avoiding the interaction between IL-1β and IL-R. It is possible to inject 150 mg or 300 mg of canakinumab every 4–8 weeks due to its half-life of 26 days [11].

Colafrancesco et al. studied four AOSD patients who were switched from anakinra to canakinumab. The results were promising after a mean of 22.1 months. After 45 months, one patient achieved remission and could discontinue therapy without relapsing during follow-up [23].

In research published in 2018, Ugurlu et al. described the use of canakinumab to treat 11 individuals with therapy-refractory AOSD. CS, csDMARDs (MTX, LEF), and bDMARDs (tocilizumab (TCZ), anakinra, infliximab (IFX), adalimumab (ADA), etanercept (ETN), and rituximab (RTX)) are prior therapies. Eight of the eleven patients were still being given canakinumab at doses of 300, 150, or 200 mg every two weeks. After just one injection, one patient experienced complete remission. Ferritin, ESR, and the global visual analog scale (VAS), as indicated by the patient, all greatly improved. Six individuals were receiving background CS of up to 10 mg of prednisolone per day at the time of the study [85].

Cavalli et al. studied the effectiveness of canakinumab as first-line biologic therapy in their case series of four patients with AOSD who were resistant to CS and MTX. All patients’ clinical characteristics and test indicators significantly improved with canakinumab. Canakinumab’s potent anti-inflammatory effects in this small case series had a notable steroid-sparing effect [99].

Campochiaro et al. showed a significant response in 6 of 10 AOSD patients treated with canakinumab. Previous therapies, such as CS, csDMARDs, and anakinra (*n* = 5), were unsuccessful. Canakinumab, 300 mg once every four weeks, led to a temporary remission of clinical and laboratory evidence of disease activity, regardless of prior therapeutic regimens. In addition, concurrent usage of Cs and csDMARDs was reduced or even terminated without relapse [85].

In the phase II, randomized, double-blind, placebo-controlled CONSIDER study (Canakinumab for Treatment of Adult-Onset Still’s Disease to Achieve Reduction of Arthritic Manifestation), a multicenter, investigator-initiated trial demonstrated that canakinumab improved several clinical aspects of AOSD while showing a favorable safety profile [85,100].

In research published in 2020, Vitale et al. analyzed data from nine AOSD patients who were given canakinumab at a dose of 150 mg every four weeks. CS, NSAIDs, csDMARDs (MTX, LEF, cyclosporine A), and biologic medicines (TCZ, ADA, ETN, and anakinra) had all previously been used to treat the patients. Canakinumab was administered as monotherapy to four patients. The majority of patients at three months were in remission. After 45 months, a long-lasting remission caused one patient to stop receiving treatment. At month 3, there was a significant overall decrease in CS in this trial. While taking canakinumab, two individuals were even able to discontinue taking their daily steroids. The same occurred in two cases with concurrent MTX medication [93].

Kedor et al. conducted a randomized, double-blind, placebo-controlled, multicenter trial to assess the effectiveness of canakinumab in the treatment of refractory AOSD with articular presentation. In the first three months, 18 individuals received canakinumab, and 17 received a placebo. Non-responders on placebo shifted to canakinumab as a rescue medication after that time. After six months of being blinded, respondents were given open-label medicine. After the 12-week period, the Disease Activity Score (DAS)28-ESR decreased by more than 1.2 in 12 individuals. At week 12, the canakinumab arm had a fever-free rate of 77.8%, while the placebo arm had a fever-free rate of 64.7% of patients. After 12 weeks, skin symptoms were comparable in both groups. Twelve canakinumab responders at week 12 were still responding at week 24, and two were still responding at week 20. Seven patients entered the long-term extension period, with three having low disease activity and four still in DAS28 remission [85].

In a study published in 2020, Tomerelli et al. revealed that 13 AOSD patients treated with canakinumab showed a strong and quick clinical response as well as a significant steroid-sparing impact with follow-ups ranging from 3 to 18 months [101].

Ugurlu et al. collected data on ten patients with AOSD treated with canakinumab and reported a case of latent tuberculosis (TB) reactivation nine months after the initial injection while receiving isoniazid chemoprophylaxis. Without being more precise, the authors linked this outcome to prior treatment exposure to numerous biologic drugs. In this trial, previous treatment included IFX, ADA, and ETN [85].

In the research by Campochiaro et al., a 69-year-old man with a systemic form of AOSD developed leucopenia following treatment with canakinumab after anakinra and TCZ failed. A 51-year-old patient from the same cohort contracted herpes zoster during therapy with canakinumab in conjunction with MTX following the failure of anakinra. Both individuals received concurrent background prednisolone therapy at a dosage of 10 mg daily [85].

In single observational center research of 13 patients with AOSD, Tomelleri et al. found three AEs: Herpes zoster reactivation, prostatitis, and moderate leucopenia [101].

Four SAEs were reported in 26 canakinumab-exposed individuals in the trial by Kedor et al. A 36-year-old female patient who had never used biologics experienced non-life-threatening transaminitis that cleared once the medication was stopped. A 51-year-old female patient had patello-femoral pain syndrome, a 30-year-old male patient developed deep vein thrombosis, and a 66-year-old female patient developed hypotonia, requiring hospitalization. The results of a liver biopsy led to the diagnosis of drug-induced hepatotoxicity. In addition, this trial documented 47 adverse events, 17 of which were non-serious infections (nasopharyngitis in the majority of cases) and 10 of which were gastrointestinal illnesses (mostly nausea) [85].

Laskari et al. [102] reported two cases of severe pneumonia in a retrospective longitudinal outcome study of 50 consecutive patients with refractory AOSD, one of which resulted in therapy discontinuation. However, no particular facts about the canakinumab dose or past biologic therapy in these individuals were provided. In all, 20% of the participants had infections, including five in the respiratory system, two in the lower urinary tract, one fungal infection in the oral cavity, one fungal infection in the external genital area, and one moderate *Staphylococcus* skin and soft tissue infection. Three patients experienced drug-induced leucopenia. The authors found that canakinumab was safe and well-tolerated by most of the patients during the 24-month follow-up period.

#### 5.3.3. Rilonacept

Rilonacept, an IL-1 tyrosine-rich amelogenin polypeptides (trap) molecule, has a longer half-life than anakinra and binds both IL-1α and IL-1β with great affinity [28]. At 160 mg/week, it can alleviate clinical symptoms and induce prolonged remission in patients with resistant AOSD. It also functions as a steroid-sparing agent [103].

Experience with rilonacept has demonstrated its efficacy in treating both arthritis and systemic symptoms in individuals with refractory AOSD [103,104].

In a 24-month follow-up trial, rilonacept was administered to five patients with refractory AOSD; three patients had significant clinical improvement [29].

In 2017, Gao and Petryna [105] reported two examples of efficient treatment with rilonacept of refractory patients with AOSD, after the failure of prednisone, MTX, and anakinra.

According to the data, IL-1 inhibition is a successful therapeutic method in AOSD resistant to standard therapy. Treatment with IL-1 Inhibitors is favorable in AOSD on various clinical and laboratory indicators, and it has a considerable steroid-sparing impact in most patients. The therapeutic effect is immediate and long-lasting.

Overall, IL-1inhibitors have a good safety profile. There have been reports of infections among adverse occurrences. Treatment with anakinra has been linked to injection-site responses and sporadic incidences of severe hepatotoxicity, which are reversible with treatment discontinuation [106].

### 5.4. IL-6 Inhibitors

The multifunctional cytokine IL-6 was initially identified in 1973. It is a key player in acute inflammation by promoting the differentiation of macrophages and cytotoxic T-cells, the chemotaxis of immune cells such as neutrophils and macrophages, the proliferation and activation of hematopoietic progenitors, the secretion of immunoglobulin (IG) by B-cells, and the production of acute-phase proteins by hepatocytes [7,107]. In the serum and pathological tissues of patients with AOSD, IL-6 concentrations are noticeably higher [26,30].

Even though IL-6 is generally recognized as a pro-inflammatory cytokine, it is considered pleiotropic due to its protective and regenerative activities based on distinct signaling pathways [108].

Given the association between disease activity and serum IL-6 levels in ASD patients [109], blocking IL-6 would be a promising treatment approach. The successful treatment of AOSD with IL-6-blocking medications supports the pathogenic role of IL-6 [53].

#### 5.4.1. Tocilizumab

TCZ represents a humanized anti-IL-6 receptor antibody that precisely blocks IL-6 by recognizing both membrane-bound and soluble versions of the IL-6 receptor [110].

In several observational studies and case reports, treatment with TCZ reduced both systemic and articular symptoms. Clinical response rates recorded vary from 64 to 100 percent [111,112,113,114].

Eleven patients completed the 6-month follow-up in the first case study of TCZ in fourteen patients with intractable AOSD at a dosage of 5–8 mg/kg every two or four weeks. Over six months, 57% of patients had complete resolution of clinical activity, and the maintenance dose of CS was lowered, suggesting that TCZ may be an effective alternative treatment for multidrug-resistant cases of AOSD [115].

TCZ performed better in a short retrospective cohort study from Japan compared to TNFα inhibitors ETN and IFX. TCZ had a continuation rate of 90.9%, much higher than IFX’s rate of 11.1% and ETN’s rate of 25% [116].

In 2018, a double-blind, placebo-controlled, randomized phase III trial was conducted to assess the efficacy and safety of TCZ. TCZ was linked to a notable improvement in systemic and articular clinical symptoms, a significant steroid-sparing benefit, and an acceptable safety profile in this research. In this study, 27 patients with AOSD refractory to CS were randomized to receive TCZ at a dose of 8 mg/kg or placebo intravenously every two weeks for 12 weeks of double-blind treatment, followed by 40 weeks of open-label treatment with TCZ. TCZ showed a substantially more potent effect on CS-sparing than the placebo did. At week 12, the dose of CS was reduced by 46.2% in the TCZ group and 21.0% in the placebo group (*p* = 0.02). Infections, AOSD aggravation, drug eruption, anaphylactic shock, and aseptic necrosis of the hips were among the SAE in the TCZ group [113]. During the trial, neither MAS nor gastrointestinal perforation was reported [117].

According to a meta-analysis by Ma et al. [78], TCZ as an adjuvant therapy induced complete remission in 77.9% of refractory cases.

Tocilizumab was utilized initially in refractory AOSD patients by Iwamoto et al. in 2002, with promising results [118]. A pilot trial conducted by Li et al. in China found that combining TCZ with csDMARDs or CS can improve clinical and laboratory symptoms of refractory AOSD patients and contribute to CS withdrawal [119]. In a case series of 14 patients with persistent AOSD, TCZ therapy resulted in complete remission of clinical disease activity in 57% of patients and a significant reduction in CS maintenance dose [28].

Additionally, specific case reports have shown that TCZ is efficient in treating MAS, pulmonary arterial hypertension, and thrombotic thrombocytopenic purpura, all systemic complications linked to AOSD [120,121].

The most frequent AEs are mild infections, injection-site reactions, neutropenia, and hepatotoxicity. Serious infections and intestinal perforation due to diverticulitis are examples of the rare but severe adverse effects of TCZ [122,123].

#### 5.4.2. Sarilumab

Sarilumab is an entirely human anti-IL-6Rα monoclonal antibody reported to be [124] effective as a steroid-sparing agent [125].

In a 25-year-old male patient with CS AOSD, sarilumab improved clinical symptoms while sparing corticoid, according to a case study [124].

The hypothesis supporting the use of sarilumab in refractory cases is based on the fact that in systemic AOSD, the high levels of IL-6 may exceed the neutralizing capacity of TCZ; therefore, direct inhibition of the IL-6 receptor may help to reduce the pro-inflammatory activity of IL-6 more thoroughly.

### 5.5. IL-18 Inhibitors

IL-18 is a pro-inflammatory cytokine of the IL-1 superfamily whose activity is controlled by the natural IL-18 binding protein (IL-18BP). Monocytes, macrophages, and dendritic cells express this IL [126]. Pro-inflammatory responses are triggered when IL-18 binds to its receptors (IL18Rα and IL-18Rβ). IL-18 levels were elevated and linked to disease activity in AOSD patients [37], and significantly higher IL-18 levels were found in AOSD patients with MAS [49].

Three IL-18 inhibitors are now being studied; however, data on the therapeutic efficacy of IL-18 blockage is still insufficient.

#### 5.5.1. Tadekinig Alpha 

Tadekinig alpha, a recombinant human IL-18BP, binds IL-18 with a strong affinity and subsequently prevents the release of TNFα, IFN-γ, and IL-1 [127].

In phase 2 multicenter European research conducted in 2018 by Gabay et al., Tadekinig alpha proved its potential efficacy and acceptable safety profile. Twenty-three patients with the long-term multidrug-resistant disease who had fever or CRP levels above ten mg/l were enlisted; 50% of them had previously had csDMARDs, and nearly a third had received prior biologic treatments. Patients were administered 80 or 160 mg of Tadekinig alpha thrice weekly. The response rate was approximately 50 percent, and the overall safety profile was favorable. Ferritin, IL-6, neutrophils, S100A8/9, and S100A12 levels fell considerably. All responders with increased IL-18 levels at baseline had undetectable levels of free IL-18 in their blood at the end of treatment.

Due to injection-site reactions, one patient in group two had to discontinue treatment one week after the experiment began. Most of the 47 drug-related AEs were skin responses, upper airway infections, and arthralgias. A volunteer aged 60 years old experienced toxic optic neuropathy, resulting in the permanent termination of the study. Overall, the safety profile of Tadekinig alpha was satisfactory [128].

Kiltz et al. published a study in which two individuals with AOSD were treated for several months with Tadekinig alpha. The first patient maintained clinical remission for two years while on a daily prednisone dose of less than 5 mg. The clinical response of the second patient, who had Tadekinig alpha for more than two years, was maintained. One patient administered 160 mg of Tadekinig alpha had an upper airway infection. However, this participant’s risk of infection was likely already elevated due to prolonged exposure to greater CS doses [129].

#### 5.5.2. AVTX 007 

AVTX 007 (formerly CERC 007, AEVI 007, and MEDI 2338) is a fully human, high-affinity anti-IL-18 monoclonal antibody, being researched for treating autoinflammatory diseases, including AOSD.

The Phase I multicenter, open-label study will involve 12 patients, six of whom will receive 7 mg/kg of AVTX-007 intravenously (iv). Based on the safety outcomes of the first cohort, six additional participants will receive a dose increase or decrease of AVTX-007. Effectiveness, safety, and tolerability will be examined [9].

#### 5.5.3. APB R3 

APB R3 is a long-acting recombinant fusion protein comprised of IL-18BP linked to an anti-human serum albumin Fab fragment via a peptide linker. An early study on APB R3 for the treatment of AOSD is being performed in South Korea as of October 2021 [9].

### 5.6. IL-17 Inhibitors

As a result of its role in neutrophil recruitment, IL-17 helps to keep the inflammatory phenotype present [130]. There has been an IL-17 elevation reported in AOSD [25].

Given the detrimental role of IL-17 in AOSD pathogenesis [50], the administration of IL-17 inhibitors to AOSD patients with a steroid-sparing effect appears plausible.

#### Secukinumab 

Secukinumab is a human monoclonal antibody that recognizes and binds to and neutralizes the interleukin-17A receptor. With a favorable safety profile, it is presently approved to treat psoriasis, AS, non-radiographic axial spondyloarthritis (nr-axSpA), and PsA.

Mitrovic et al. [131] described a single case of an AOSD patient who obtained complete remission with secukinumab, after the loss of efficacy to anakinra and MTX following the development of SpA.

### 5.7. TNFα Inhibitors

TNFα, a member of the TNF superfamily, is primarily produced by lymphocytes and activated macrophages [132]. It can trigger various cellular and molecular behaviors and events by binding to two receptors, TNFRI and TNFRII [133]. Serum and synovial membrane TNFα levels are markedly higher in individuals with systemic or chronic AOSD [30]. The efficacy of TNFα blockers in AOSD is controversial [134].

IFX, ETN, and ADA were the first TNFα inhibitors to be utilized in single case reports or short series in the early 2000s [135].

Essential data on the safety and efficacy of anti-TNFα drugs are provided by the study conducted by Fautrel et al., which involved the administration of IFX or ETN to 20 patients with AOSD (five with systemic and fifteen with polyarticular forms), whose response to MTX and CS was inadequate. Four individuals treated with IFX obtained complete remission, whereas nine patients achieved partial remission. With ETN therapy, the majority of patients obtained a partial response, while just one patient reached complete remission.

#### 5.7.1. Infliximab

IFX was initially administered to three individuals with chronic and aggressive AOSD in 2001, and it showed long-term success in the treatment of refractory AOSD patients [29].

Kraetsch et al. reported in 2001 that IFX therapy resulted in significant improvements in clinical symptoms and normalization of laboratory parameters in all six AOSD patients with severe disease activity [29,37].

Dechant et al. (2004) indicated that 87.5% of patients treated with IFX for eight cases with multidrug-resistant AOSD responded. Five of these patients stayed in remission after IFX was discontinued, and one of them was transferred to ETN owing to infusion problems. Only one responder and one non-responder to these biological agents required chronic treatment [136].

In a 2018 evidence-based review, Zhou et al. showed that TNFα inhibitors might not be beneficial in treating AOSD [114].

#### 5.7.2. Etanercept

ETN is a soluble recombinant version of the human TNFα-receptor fusion protein of 75 kDa. In 2002, 12 AOSD patients with active arthritis resistant to csDMARDs were included in open-label research conducted by Husni et al. As a result, arthritis improved in seven patients, with non-significant AEs [137]. 

#### 5.7.3. Adalimumab

ADA’s safety and efficacy in AOSD remain uncertain due to small sample numbers and a lack of relevant studies [138].

### 5.8. IFN- γ Inhibitors

Given the pathogenic role of IFN-γ in AOSD [37], its blockade may effectively treat AOSD with or without concomitant MAS [139]. 

#### Emapalumab 

Emapalumab is a fully human monoclonal antibody that inhibits receptor dimerization and signal transduction to neutralize both free and receptor-bound IFN-γ [140]. Gabr et al. [141] found that emapalumab significantly reduced fever and enhanced laboratory results in one patient with AOSD exacerbated by MAS.

### 5.9. Janus Kinases Inhibitors

Both type I and type II cytokine receptors bind to the Janus kinases (JAKs), including TYK2, JAK1, JAK2, and JAK3 [142]. Upon binding to their receptors, several cytokines induce additional inflammatory gene expression via JAK pathways, which increase the inflammatory signaling loop.

JAK inhibitors have been a viable therapeutic method for treating inflammatory conditions, due to their pronounced effects on cytokine generation and immune response modulation [143]. JAK inhibitors reduce the effects of IL-6, IL-10, IFN-γ, and Granulocyte-Macrophage Colony-Stimulating Factor (GM-CSF), which are substantially implicated in the pathophysiology of AOSD.

Data on the efficacy and safety of JAK inhibitors in treating AOSD are yet restricted to case studies. 

#### 5.9.1. Baricitinib

For the first time in 2019, the effectiveness of baricitinib was described in a VCS-dependent refractory AOSD patient. In another study, this outcome was confirmed by Gillard et al. [144].

According to Kacar et al., baricitinib successfully treated two AOSD patients who failed to respond to both biological therapy and csDMARDs. A patient with refractory AOSD was successfully treated with baricitinib and anakinra [145].

#### 5.9.2. Tofacitinib

In a study by Hu et al., 14 people with refractory AOSD were given tofacitinib. Seven had a complete remission, six had a partial response, and one relapsed after their prednisone dose was lowered. The results of this study imply that tofacitinib could be an option for treating AOSD, particularly the arthritic variant. Furthermore, tofacitinib [146] reduced the requirement for steroid medication. Another case report [147] describes the effective treatment of AOSD with tofacitinib in a female HIV-positive patient.

#### 5.9.3. Ruxolitinib

In experimental murine models of Hemophagocytic lymphohistiocytosis (HLH), the JAK1/2 inhibitor ruxolitinib is known to drastically lower the proliferation and activation of immune modifying IFN-γ and other cytokines [148].

After 28 days of oral therapy, 12 children with secondary HLH demonstrated clinical improvement [149]. Additionally, two AOSD patients taking steroids had partial responses [144].

### 5.10. GM-CSF Inhibitors

After GM-CSF binds to GM-CSF receptor a (GM-CSFRa), macrophage and neutrophil quantity and function are increased in inflammatory lesions, resulting in the excessive secretion of pro-inflammatory cytokines, such as IL-1β, IL-6, IL-12, IL-23, and TNFα [150,151,152].

#### Mavrilimumab and Otilimab 

Mavrilimumab (CAM-3001), an IgG4 mAb that inhibits GM-CSFRa directly, and Otilimab (MOR-103), a GM-CSF-binding IgG1 mAb, prevent the release of pro-inflammatory cytokines.

In 2 phase IIb studies and an open-label extension study involving a total of 442 RA patients, 65% of patients achieved remission using DAS28-CRP.

Furthermore, mavrilimumab was recently used to treat severe COVID-19 pneumonia based on the idea that inhibition of GM-CSF could reduce the hyperinflammation caused by the virus [153]. As a result, these inhibitors could be employed to treat both the chronic articular pattern and the systemic type of AOSD.

### 5.11. NLRP3 Inflammasome Inhibitors

The inflammasome NLRP3 and its components are known to play a role in autoinflammatory processes. So far, various agents capable of binding NLRP3 and hence causing IL-1 release have been explored.

#### Dapansutrile 

Dapansutrile (OLT1177), an orally active beta-sulfonyl nitrile, acts as a direct inhibitor of NLRP3 and is now being studied in the treatment of gout [154]. Oral administration of varying doses of dapansutrile resulted in a considerable reduction in joint pain and swelling. In addition, a decrease in pro-inflammatory cytokines, particularly IL-6, was seen [155].

Despite the limitations of this investigation, the drug’s efficacy, strong safety record, and oral administration make it a potentially effective treatment for gouty patients and those with AOSD, particularly in those with primarily articular involvement.

### 5.12. Long Non-Coding RNAs 

Long noncoding RNAs (lncRNAs) are regarded as essential immune response regulators [156,157,158,159]. Their expression is linked to specific pathways or cytokines that contribute to the pathophysiology of AOSD.

Blood levels of NEAT-1 (nuclear enriched abundant transcript 1) in patients with AOSD correlate considerably with the expression of other lncRNAs following treatment with cyclosporine or anti-IL-6. 

The myocardial infarction-associated transcript (MIAT) was also reported to inhibit IL-1β and TNFα, whereas THRIL (TNFα and hnRNPL-related immunoregulatory lncRNA) was discovered to increase TNFα. Increased MIAT levels and decreased THRIL expression were seen in AOSD compared to controls [160].

According to the different types of lncRNA signatures found, it may be feasible to understand the axis or set of cytokines primarily engaged in the pathogenesis of AOSD, hence contributing to the treat-to-target approach and effective patient management.

### 5.13. Other Therapeutical Approaches 

#### 5.13.1. Rituximab

RTX is a chimeric anti-CD20 monoclonal antibody that can block T cell activation and the generation of pro-inflammatory cytokines [161]. It is authorized for the treatment of RA. However, only a few case studies [162,163] have emphasized the possible effectiveness of RTX (375 mg/m^2^ given twice at 4-week intervals) in refractory AOSD patients.

#### 5.13.2. Abatacept

Abatacept (CTLA4IgFc) is a co-stimulation modulator that reduces T-cell activation by binding to CD80 and CD86 receptors on antigen-presenting cells (APCs) and blocking their interaction with the CD28 receptor on T lymphocytes. According to studies, Abatacept may be useful in AOSD patients resistant to csDMARDs, anti-TNFα medications, and even IL-1 receptor antagonists [164]. 

#### 5.13.3. IVIGs

In two open-label trials, intravenous immunoglobulins (IVIG) were shown to be efficacious and well-tolerated in half of the patients when administered at the typical dose of 2 g/kg over 2–5 days each month [165], though they should be used in certain circumstances or when life-threatening symptoms appear.

Figure 5 highlights a brief summary of anti-cytokine therapies studied so far in treating AOSD.

### 5.14. MAS Treatment

MAS is an uncommon, potentially fatal inflammatory condition. The mortality rate approximates 41%. Secondary MAS affects up to 15% of AOSD patients [166]. It is regarded as the disease’s most serious consequence, manifesting clinically as severe hyperinflammation, pancytopenia, liver damage, markedly elevated ferritin, and coagulopathy [167,168].

MAS is caused by increased macrophage activation and proliferation, T lymphocytes, pro-inflammatory cytokine hypersecretion, tissue infiltration, haemophagocytosis, and tissue destruction. The pathophysiological mechanism of MAS suggests a “cytokine storm” of IL-1β, IL-2, IL-6, IL-18, IFN-γ, M-CSF, sTNFα-R, and IL-1R antagonist (IL-1Ra) [18,169].

AOSD is less usually related to MAS and is typically treated with TNFα, IL-1, IL-6, or IL-18 inhibitors [1]. Although higher IFN-γ levels in AOSD patients have been documented [141], this cytokine has not been evaluated for therapeutic intervention.

In the pathogenesis of AOSD, IL-1β has been identified as a critical inflammatory mediator. Multiple data suggest that anakinra can be helpful for individuals with AOSD who develop MAS [170,171], and the mechanism involves the suppression of pro-IL-18 transforming into an active cytokine [42].

In AOSD, unresponsive to conventional therapy and other biologics, the suppression of IL-1 is an effective therapeutic strategy. Consensus treatment recommendations advise using IL-1 inhibitors as early as possible in systemic forms of AOSD as the first line of biologic treatment and AOSD-related MAS as both the first and second line of biologic therapy [23].

In 2016 research by Watanabe et al. [120], TCZ was helpful for AOSD patients with MAS. Nonetheless, MAS developed following TCZ treatment in one patient with refractory AOSD, indicating that caution should be maintained in the highly active state of this illness [172]. So, even though IL-6 inhibition was successful in treating AOSD, there is a worry that it may cause MAS, mainly based on the experience of systemic juvenile idiopathic arthritis [173,174].

In the study of TCZ in AOSD published by Naniwa in 2021, no cases of MAS were identified, even though multiple other case reports [175] have shown the likelihood of TCZ causing this complication.

However, the association between IL-6 inhibition and MAS in AOSD remains unknown, and more cases and research are needed to clarify it.

As we have previously specified, in a study by Gabr et al. [141], one patient with AOSD and MAS treated with emapalumab had considerably decreased fever and improved laboratory values.

## 6. Conclusions

AOSD continues to be a complicated and diverse illness. As an uncommon condition, AOSD is tough to cure, but much more difficult to diagnose. Before an appropriate diagnosis and efficient treatment plan can be adopted, AOSD patients must typically undergo a journey marked by confusing symptoms, misdiagnosis or delayed diagnosis, and a series of ineffectual treatments. This delay in diagnosis might result in a longer hospital stay and a greater financial burden for the patient. In addition, it may precipitate the onset of uncommon and possibly fatal AOSD complications. Additionally, a better understanding of the etiology of AOSD is essential.

In milder forms of AOSD, NSAIDs alone may be adequate to manage illness symptoms; however, CS therapy is usually necessary for moderate and severe cases. To achieve steroid-free remission in CS-dependent patients and to manage refractory instances of illness, corticoid-sparing medicines are typically required due to the substantial toxicity burden associated with continuous steroid therapy. Commonly used csDMARDs include MTX, cyclosporine A, and LEF. 

Improvements in AOSD patients’ quality of life and ability to cope have resulted largely from recent developments in the development of biological drugs. Anti-cytokine drugs are an effective and safe pharmacological alternative to csDMARDs and the only viable treatment choice in the most severe and refractory patients. Specific suppression of IL-1 and IL-6 is currently recognized as a safe and effective medication for illness management. Anakinra, canakinumab, and TCZ successfully achieve clinical and biochemical remission in many AOSD patients, and they have a significant steroid-sparing impact. Canakinumab is currently the only FDA-approved medicine for AOSD in the United States, but both canakinumab and anakinra are approved for AOSD in Europe.

Although our understanding of AOSD has increased over the past decade, there are still significant gaps in our knowledge of its diagnosis, the most helpful biomarkers, and the treatment strategy. An accurate picture of the AOSD burden is required to guide healthcare actions and initiatives. Due to the difficulty of performing large-scale prospective studies in the setting of uncommon illnesses, countrywide registries and high-quality RCTs with a smaller number of patients can assist in bridging the remaining knowledge gap.

## Figures and Tables

**Figure 1 ijms-23-12810-f001:**
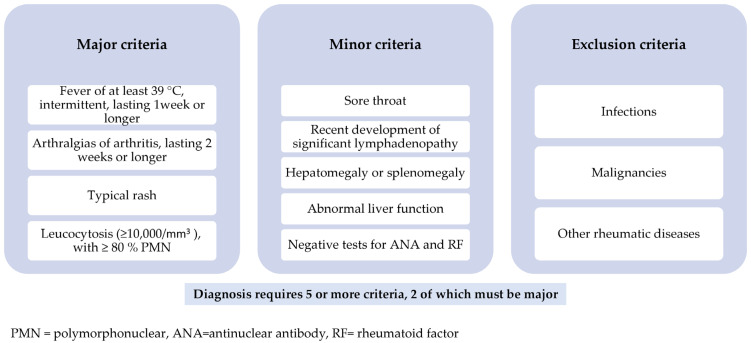
Yamaguchi criteria for AOSD (1992).

**Figure 2 ijms-23-12810-f002:**
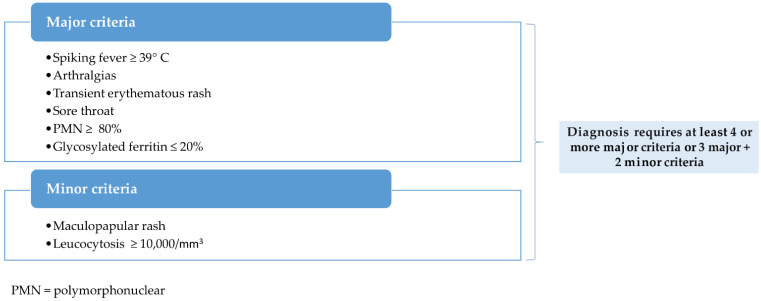
Fautrel criteria for AOSD (2002).

**Figure 3 ijms-23-12810-f003:**
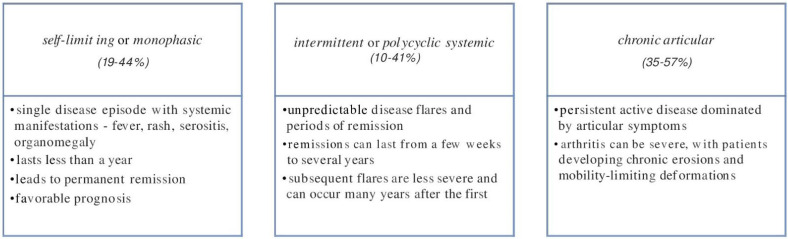
AOSD evolution patterns.

**Figure 4 ijms-23-12810-f004:**
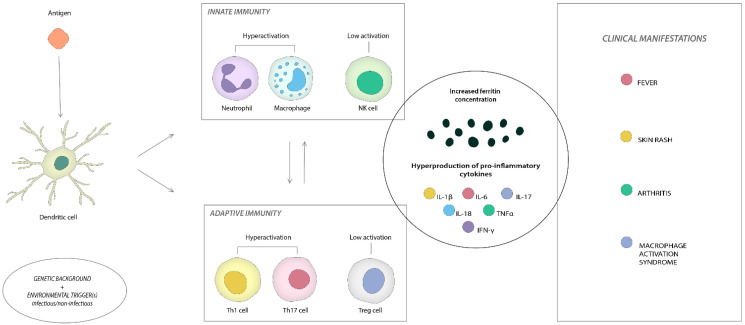
AOSD pathogenesis.

**Figure 5 ijms-23-12810-f005:**
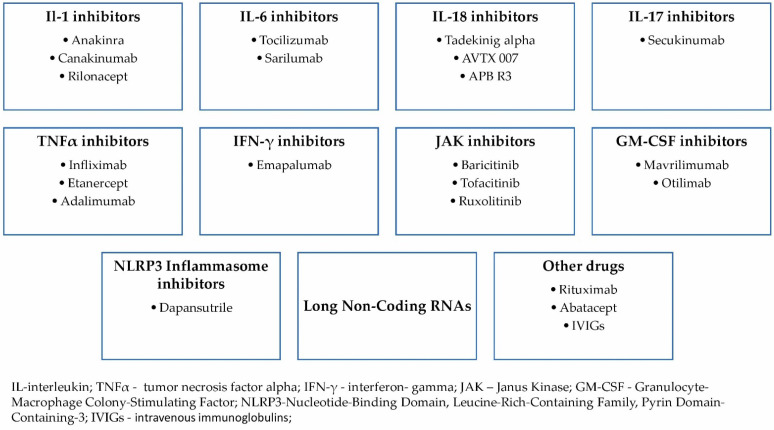
Anti-cytokine therapy in AOSD.

## Data Availability

Not applicable.

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
