# Peer review of "Adult-Onset Still’s Disease—A Complex Disease, a Challenging Treatment"

_ijms, 2022, doi:10.3390/ijms232112810_

Round 1

Reviewer 1 Report

The authors Macovei et al. reviewed the clinical, diagnostic, pathophysiological and therapeutic aspects of adult Still’s disease. The authors should provide a better manuscript structure and not just cite facts. Please find some examples below.

Major comments

1. Title: ‘’Adult-onset Still disease - a complex disease, a promising treatment’’. Please correct to Still’s disease. The title is bit confusing. The phrase: A treatment challenge? or something similar may be more appropriate.

2. Introduction: Please provide first the information on the disease incidence, then refer to the clinical and laboratory findings.

3. Clinical picture and diagnostic criteria: It is not clear to which criteria the authors exactly refer when they mention about sensitivity, specificity and exclusion criteria. Also, the prognosis and MAS should be reviewed in separate paragraphs.

4. Pathogenesis: a. A more condensed text would be easier to read.

b. “Some clinical signs of AOSD, such as fever spikes, lymphadenitis, and elevated liver enzymes, resemble viral or bacterial infections, suggesting that infection may initiate the inflammatory response in AOSD.”: Is this an evidence-based observation?

c. I would suggest to avoid absolute statements such as: ‘’IL-1β is the characteristic pro-inflammatory cytokine.” or ”Increasing concentrations of IL-1β cause the entire range of AOSD symptoms”.

5. Treatment: In general, please try to bring together the results of the available studies and not just describe the main results of each paper.

a. Avoid repetitions such as in paragraph 4.2. NSAIDs, CS and csDMARDs treatment or at the end of paragraph 4.2.2 CS.

b. ‘’CS are indicated for patients with more severe symptoms’’: not necessarily, since CS are the first-line treatment.

c. “Several AOSD patients, however, experience illness flares or become steroid-dependent after tapering or interruption”: How many?

Author Response

Thank you for the very interesting and useful suggestions.

Point 1

Title: ‘’Adult-onset Still disease - a complex disease, a promising treatment’’. Please correct to Still’s disease. The title is bit confusing. The phrase: A treatment challenge? or something similar may be more appropriate.

Response 1:

Thank you for the recommendations regarding the title of the article. We changed it as you proposed.

Point 2

Introduction: Please provide first the information on the disease incidence, then refer to the clinical and laboratory findings.

Response 2:

The authors made the change.

Point 3

Clinical picture and diagnostic criteria: It is not clear to which criteria the authors exactly refer when they mention about sensitivity, specificity and exclusion criteria. Also, the prognosis and MAS should be reviewed in separate paragraphs.

Response 3:

We made the corrections.

Point 4

Pathogenesis:

   a. A more condensed text would be easier to read.

   b. “Some clinical signs of AOSD, such as fever spikes, lymphadenitis, and elevated liver enzymes, resemble viral or bacterial infections, suggesting that infection may initiate the inflammatory response in AOSD.”: Is this an evidence-based observation?

   c.  I would suggest to avoid absolute statements such as: ‘’IL-1β is the characteristic pro-inflammatory cytokine.” or ”Increasing concentrations of IL-1β cause the entire range of AOSD symptoms”.

Response 4:

  a. We understand that the text is larger, but the pathogenesis is very complex; the authors appreciated the fact that by reducing this chapter, the treatment would not be understood either.

  b. Indeed, there are a number of studies that raise the hypothesis of the involvement of bacterial and viral triggers and the infections they cause in the onset of the disease.

  c. We made the necessary changes.

Point 5

Treatment: In general, please try to bring together the results of the available studies and not just describe the main results of each paper.

  a. Avoid repetitions such as in paragraph 4.2. NSAIDs, CS and csDMARDs treatment or at the end of paragraph 4.2.2 CS.

  b. ‘’CS are indicated for patients with more severe symptoms’’: not necessarily, since CS are the first-line treatment.

  c. “Several AOSD patients, however, experience illness flares or become steroid-dependent after tapering or interruption”: How many?

Response 5:

  a. We made the corrections according to the indications.

  b. We made the change and deleted the phrase. Thank you for your attention.

  c. We have not found any precise data on this in the literature.

Reviewer 2 Report

This is an excellent review on current concepts of pathology and treatment of adult onset still's disease. Despite extensive content, this is very readable. References are clear and relevant. I am not sure that Figure 2 is needed as this reiterates the headings in the text and does not add anything to the paper. 

Perhaps a figure outlining the pathological pathways with the addition of where each treatment acts may be more valuable?

Author Response

Thank you for the very interesting and useful suggestions.

This is an excellent review on current concepts of pathology and treatment of adult onset still's disease. Despite extensive content, this is very readable. References are clear and relevant. I am not sure that Figure 2 is needed as this reiterates the headings in the text and does not add anything to the paper. 

Response:

Thank you for your kind words of appreciation.

Figure 2 briefly highlights the anti-cytokine therapy described in the article. The authors considered it appropriate to make this figure in order to help the reader better retain the complex treatment of the disease.

Perhaps a figure outlining the pathological pathways with the addition of where each treatment acts may be more valuable?

Response:

This is an excellent idea and we are grateful for it. Such an image would have been particularly interesting, with cytokines involved in AOSD pathogenesis and therapies addressing their inhibition. Unfortunately, we have little time to make a suggestive figure, in which the pathogenic picture intersects with the treatment. We will take this into account in our next review.